# Gender Differences in the Relationship Between Health Literacy and Stress Among Caregivers of Older Adults with Dementia

**DOI:** 10.3390/healthcare13233064

**Published:** 2025-11-26

**Authors:** Chiara Lorini, Rita Manuela Bruno, Enrico Mossello, Yari Longobucco, Primo Buscemi, Annamaria Schirripa, Barbara Giammarco, Giuseppe Albora, Duccio Giorgetti, Massimiliano Alberto Biamonte, Letizia Fattorini, Gemma Giusti, Lisa Rigon, Giulia Rivasi, Andrea Ungar, Guglielmo Bonaccorsi

**Affiliations:** 1Department of Health Science, University of Florence, Viale GB Morgagni 48, 50134 Florence, Italy; yari.longobucco@unifi.it (Y.L.); gemyge13@gmail.com (G.G.); guglielmo.bonaccorsi@unifi.it (G.B.); 2Health Literacy Laboratory, Department of Health Science, University of Florence, Viale GB Morgagni 48, 50134 Florence, Italy; 3Medical Specialization School in Geriatric Medicine, University of Florence, Largo Brambilla 3, 50134 Florence, Italy; ritamanuela.bruno@unifi.it; 4Division of Geriatric and Intensive Care Medicine, Careggi Hospital, Largo Brambilla 3, 50134 Florence, Italy; enrico.mossello@unifi.it (E.M.); giulia.rivasi@unifi.it (G.R.); andrea.ungar@unifi.it (A.U.); 5Department of Experimental and Clinical Medicine, University of Florence, Largo Brambilla 3, 50134 Florence, Italy; 6Medical Specialization School in Hygiene and Preventive Medicine, University of Florence, Viale GB Morgagni 48, 50134 Florence, Italy; primo.buscemi@unifi.it (P.B.); annamaria.schirripa@unifi.it (A.S.); barbara.giammarco@unifi.it (B.G.); giuseppe.albora@unifi.it (G.A.); duccio.giorgetti@unifi.it (D.G.); massi.biamonte@gmail.com (M.A.B.); letizia.fattorini@unifi.it (L.F.); lisa.rigon@unifi.it (L.R.)

**Keywords:** health literacy, older adults, informal caregiver, stress, home services, dementia

## Abstract

**Background/Objectives:** This study aims to investigate the association between health literacy (HL) and stress among family caregivers of older adults with dementia. **Methods:** Older adults and their caregivers were recruited from the geriatric outpatient memory clinic of an Italian hospital. Caregiver stress was assessed using the General Health Questionnaire-12 items (GHQ-12). HL was measured using the Newest Vital Sign (NVS) and the Short Form of the Test of Functional Health Literacy in Adults (S-TOFHLA). **Results:** A total of 170 (71% females) caregivers, including spouses and offspring, were included in the analysis. According to the NVS, 53% demonstrated adequate HL, while 83% achieved adequate scores on the S-TOFHLA. The median GHQ-12 score was 15, with 48% presenting a score above 14, indicating higher stress levels; women reported significantly higher GHQ-12 scores than men. In a multivariate linear regression analysis adjusted for sex, education, and number of care tasks provided, the S-TOFHLA score showed a borderline association with the GHQ-12 score (B = −1.45; *p* = 0.064). When characteristics of the care-recipient were added to the model, the S-TOFHLA score emerged as an independent predictor of the GHQ-12 score (B = −1.41; *p* = 0.048), along with female caregiver sex and behavioral and psychological symptoms in the care-recipients. Exploratory analysis suggested that the association between HL and stress was present among male but not female caregivers. **Conclusions:** HL was associated with psychological stress in caregivers of older adults with dementia, with the relationship appearing more pronounced among male caregivers.

## 1. Introduction

In recent decades, high-income countries have experienced an epidemiological transition, leading to a marked increase in the number of older adults, many of whom live with multiple noncommunicable diseases. Among these, dementia has emerged as a leading cause of disability and dependency worldwide, impairing memory, cognitive functions, and behavior, and consequently limiting daily functioning. Dementia leads to substantial human costs on individuals, families, and societies [1].

Both formal (paid) and informal (family or friends) caregivers play a crucial role in supporting older adults with dementia at home. Their responsibilities span a wide range of tasks—from assisting with daily domestic and personal care activities to managing medications, communicating with healthcare providers, and navigating the healthcare system [1,2,3]. This issue is particularly relevant in countries such in Italy, where the aging population has led to a rise in disability, with over 90% of older adults cared for at home [4]. Caregiving deeply affects the quality of life of people with dementia, and, in turn, exerts significant effects on mental and physical health [5]. Several studies have shown that prolonged caregiving is associated with negative health outcomes, including psychological distress, increased cardiovascular risk, reduced quality of life, and even higher mortality rates among caregivers [6,7,8,9,10,11,12]. “Stress” is a multifaced process comprising stressors (i.e., challenging events), mediators (i.e., constructs that shape how threats are perceived and managed), and the stress response itself (i.e., the physical and emotional reactions elicited by a stressor). The term is often used to encompass both the stressors and resulting responses [13].

Health literacy (HL) “entails people’ s knowledge, motivation and competences to access, understand, appraise, and apply health information in order to make judgments and take decisions in everyday life concerning healthcare, disease prevention and health promotion to maintain or improve quality of life during the life course” [14]. Being health-literate involves placing one’s own health—and that of one’s family—within its broader context, understanding the factors that influence it, and knowing how to effectively address them. Individuals with adequate HL have the ability to take responsibility for their own health as well as the health of their family members [14]. From this perspective, health literacy skills can enable caregivers to provide more appropriate and effective care for the individuals they support [15].

Studies involving different types of caregivers of patients with various illnesses or conditions suggest that higher levels of HL may positively influence caregivers’ quality of life, anxiety levels, and psychological well-being [16,17,18]. However, the limited number of studies, heterogeneity in the study context, and care-recipient characteristics currently prevent definitive conclusions. According to Zhao’s theoretical perspective [16], caregivers who face health-related information demands—such as assisting patients in managing their condition—may experience a greater caregiving burden if their HL is insufficient to meet the cognitive and practical challenges of caregiving. From this perspective, the number and type of home care tasks provided could also be associated with caregiver stress.

Giving these premises, this study aims to examine the relationship between HL, caregiving tasks, and stress among family caregivers of older adults with dementia, with a focus on sex differences.

## 2. Materials and Methods

This study is part of a larger project that aimed to collect and describe the demographic, social, and health characteristics of caregivers of older adults with dementia. The project includes an initial cross sectional phase, followed by a longitudinal phase (not presented in this article). This study was approved by the Ethics Committee of the Local Health Authority responsible with territorial jurisdiction (CEAV 13592_oss) and conducted in accordance with the principles expressed in the Declaration of Helsinki. The present analysis focuses exclusively on data concerning stress among informal caregivers, who represent the majority of the study sample. Previous findings of the broader project have been reported elsewhere [19]. Specifically, a previous analysis focused on describing HL levels among caregivers and on evaluating predictors of HL [19]; regarding the latter, some characteristics of caregivers (sociodemographic data, cohabitation and kinship with the person with dementia, number of weekly hours devoted to assistance, cognitive impairment) and of the care-recipient were included. The analysis presented in this paper is therefore innovative compared to the previous one, as it focuses on caregiver stress (which was not investigated in the previous study) and on the factors associated with it.

### 2.1. Study Design

We enrolled older adults and their informal caregivers from the geriatric outpatient memory clinic of an Italian Teaching Hospital. Participation required written informed consent from both the caregivers and the older adult. In cases where the older adult was unable to provide consent due to severe dementia, enrolment was permitted based on the clinician’s judgment, in accordance with the approval from the Ethics Committee. Eligible older adults met the following inclusion criteria: aged over 64 years, attending the outpatient clinic for cognitive disorders at the Teaching Hospital, diagnosed with dementia according to the National Institute on Aging criteria, and scoring ≥4 on the Global Deterioration Scale (GDS) [20,21]. Individuals without a family caregiver and those not accompanied by their caregiver at the time of the visit were excluded. Caregivers were eligible if they provided at least four hours of care per week to an older adult with dementia.

Recruitment took place between August 2018 and March 2021. The process was prolonged due to the suspension of outpatient visits during the COVID-19 pandemic and proceeded at a rate of approximately one to two sessions per week. Based on data from a previous survey of caregivers of older patients in the same geographical area, with values expected to indicate low HL prevalence and considering a confidence interval of 95% and a margin of error equal to 0.05, we defined approximately 180 caregivers as the target sample size [19]. Specifically, the sample size was calculated with respect to the primary outcome of the general research project, which was the percentage of caregivers with a low level of health literacy, measured using two instruments (S-TOHFLA and NVS). The following formula was used:*N* = *Z*^2^*x P* (1 − *P*)*d^2^*
where

*Z*: the Z-score that corresponds to the desired confidence level;

*P*: the expected prevalence (or proportion) of the condition in the population.

*d*: the desired margin of error (or precision) for the estimate.

The demographic and clinical characteristics of the care-recipients are described elsewhere [19].

### 2.2. Caregivers’ Assessment

Regarding caregivers, data were collected on socio-demographic characteristics (date of birth, sex, nationality, mother tongue, level of understanding of the Italian language for persons coming from abroad, years of education), the number of hours devoted to caregiving, cohabitation with the care-recipient, and the specific caregiving tasks performed.

The Newest Vital Sign (NVS) and the Short Form of the Test of Functional Health Literacy in Adults (S-TOFHLA) have been used to measure caregivers’ HL. Both tools have been adapted and validated for use in Italian populations, demonstrating good face and content validity, as well as satisfactory internal consistency in previous studies (Chronbach’s alpha coefficient was 0.96 for the S-TOFHLA and 0.74 for the NVS) [22,23,24,25,26,27,28,29,30,31]. These instruments evaluate functional HL, that is the reading, writing, and numeracy skills required to effectively manage everyday health-related situations [32].

Assessment of the stress level—the primary objective of this study—was conducted using the General Health Questionnaire (GHQ). Developed by David Goldberg in the 1970s, the GHQ is one of the most widely used standardized tool for measuring emotional distress in epidemiological studies. Specifically, we employed the 12-item short version (GHQ-12), which explores the four core dimensions of distress: depression, anxiety, social deterioration, and somatic symptoms [33,34,35,36]. The GHQ-12 consists of six ‘positive’ and six ‘negative’ items, both scored so that higher values indicate greater distress. While the traditional scoring method dichotomizes responses (0-0-1-1), we applied the Likert scoring approach, which assigned ordinal values (0-1-2-3) to each response, yielding a total score ranging from 0 to 36. A cut-off corresponding to 13/14 was used to identify participants with clinically significant distress [35]. The Italian version of the GHQ-12 has demonstrated good concurrent validity with respect to the Clinical Interview Schedule, showing high sensitivity (71–75%) and specificity (73–76%) with respect to the long GHQ version, as well as satisfactory reliability [36].

Data on caregivers were collected by the members of the research team through face-to-face interviews, with the exception of the S-TOHFLA, which was self-administered.

### 2.3. Older Adults’ Assessment

Cognitive impairment was assessed using the Mini-Mental State Examination, including correction for Italian norms of age and education (MMSEc, range of the score: 0–30, with higher scores indicating better performance) [37]. Behavioral and psychological symptoms associated with dementia were evaluated using the Neuropsychiatric Inventory (NPI), a 12-item scale that measures the frequency and severity of behavioral symptoms over the previous four weeks, based on caregiver reports. For each reported symptom, a domain score is obtained by multiplying its frequency (1 to 4) by its severity (1–3). The total NPI score corresponded to the sum of the domain scores [38,39].

### 2.4. Sample Description

A total of 180 informal caregivers were enrolled in this study. For the purposes of this analysis, only spouses and offspring were considered (*n* = 170; median age: 59 years; 71% females).

Table 1 reports the caregivers’ main characteristics. Forty-four percent lived with the care-recipient, and the median weekly number of care hours was 30. The median educational level was 13 years. Half of the caregivers performed five or more different care tasks. The majority provided assistance with medication (90%), supervision (89%), household chores (75%), and personal hygiene (65%). Thirty-seven percent had received some form of healthcare training.

The median NVS score was 4; 53% showed adequate HL, 26% had a possibility of limited HL, and 21% showed a high likelihood of limited HL. The median S-TOFHLA score was 94, with most participants (83%) showing adequate HL, while 17% had inadequate or marginal HL. The median GHQ-12 score was 15, with 48% of the caregivers scoring above 14.

Compared with men, female caregivers were more frequently offspring (79% vs. 56%), younger (mean age: 60.3 vs. 67.5 years), and responsible for a greater number of care tasks (mean: 4.7 vs. 4.1) (Table 2).

### 2.5. Statistical Analysis

All analyses were performed using IBM SPSS 29.0. Categorical variables were reported as frequencies and percentages, while continuous variables were summarized as the mean (SD) and median (IQR). Bivariate analyses were conducted using the Chi2 test, Student’s *t*-test, the ANOVA, and Spearman correlation analysis, as appropriate. Normality of residual distributions was assessed using P-P plots, and homoscedasticity was evaluated for *t*-tests and ANOVA measures. The effect size for mean comparisons was estimated with Cohen’s d for the Student *t*-test and eta-squared for ANOVA.

Variables significantly associated or correlated with the GHQ-12 scores at *p* < 0.10 in the bivariate analyses were included as independent variables in a linear regression model (backward deletion method), with the GHQ-12 score as dependent variable. Two models were performed: Model 1, including caregivers’ characteristics only, and Model 2, including both caregivers’ and patients’ characteristics.

Regression assumptions were tested as follows: normality of residuals was verified using the P-P plots, collinearity was assessed using the Variance Inflation Factor (VIF), with VIF < 5 indicating acceptable levels, and homoscedasticity was evaluated by inspecting scatterplots of standardized residuals by standardized predicted values and confirmed with the Breusch–Pagan test.

A post hoc analysis was conducted to explore the association between HL and stress within caregivers of different sexes.

For all analyses, a *p*-value < 0.05 was considered significant.

Missing values were excluded by listwise deletion.

## 3. Results

### Caregivers’ Features and Psychological Stress

A higher GHQ-12 score was significantly associated with female sex (mean value: females = 15.8 ± 4; males = 14.2 ± 2.7; *p* = 0.004), as well as with lower HL as measured by the S-TOFHLA (mean value: inadequate or marginal HL = 16.6 ± 3.6; males = 15.0 ± 3.7; *p* = 0.040), while the association with NVS scores was not significant (Table 3).

The GHQ-12 score was also positively significantly correlated with the number of tasks performed (Rho = 0.192; *p* = 0.013) (Table 3).

Furthermore, GHQ-12 scores were significantly correlated with several care-recipient characteristics: the lower the MMSE score, the higher the GHQ-12 score (Rho: −0.186; *p* = 0.017); the higher the NPI score, the higher the GHQ-12 score (Rho: 0.400; *p* < 0.001).

The final models obtained using backward deletion are described in Table 4. In Model 1, which included caregivers’ characteristics only, both the S-TOFHLA score and the number of care tasks performed presented an association with caregiver stress that approached statistical significance, while female sex showed an independent association. In particular, female sex and the number of tasks performed were positively associated with the GHQ-12 score (for female sex: B = 1.32; *p* = 0.043; for the number of tasks provided: B = 0.31; *p* = 0.060), whereas the S-TOFHLA score was negatively associated with the GHQ-12 score (B = −1.45; *p* = 0.064). However, due to the lack of homoscedasticity, the estimates of Model 1 may not be considered stable (see Appendix A). When the care-recipients’ features were also included (Model 2), female sex and the S-TOFHLA score remained independent predictors of the GHQ-12 score (female sex: B = 1.72; *p* = 0.004; S-TOHFLA: B = −1.41; *p* = 0.048). Among the care-recipients’ features, only the NPI score significantly and positively predicted caregiver stress (B = 0.15; *p* < 0.001). Model 2 provided robust estimates across all predicted values, as confirmed by the assessment of homoscedasticity (see Appendix A).

In the exploratory analysis stratified by caregiver sex, a significant association emerged between HL and GHQ-12 scores among males: lower GHQ-12 scores were associated with higher HL, both when assessed using the three-level NVS [eta-squared 0.15 (95% CI 0.00, 0.32)] and the S-TOFHLA [eta-squared 0.14 (95% CI 0.01, 0.33), corresponding to a medium-to large effect size]. Conversely, among female caregivers, no significant association emerged between HL and the GHQ-12 score (Figure 1).

## 4. Discussion

Informal caregivers are essential partners in planning and providing home care tailored to the needs of a person with dementia. However, caregiving for these people is a complex and demanding task that can substantially affect caregivers’ physical and psychological well-being [12,40,41,42,43,44,45,46]. Our findings support this view. Stress levels among caregivers of patients with dementia are both high and common, as approximately half of the sample scored above 14 on the GHQ-12, indicating a need for psychological support. In this context, investigating factors related to caregivers’ stress is a priority to inform interventions aimed at protecting the health of both care-recipients and their caregivers.

Recently, HL has gained increasing attention as a determinant of health, not only from an individual perspective but also as a collective attribute of families, social networks, communities, and populations [47,48,49,50]. In this broader view, it is also understood as an individual capacity not only to manage one’s own health but also to care for others, such as family members or patients. When the care-recipients are individuals with dementia, informal caregivers play a pivotal role as a health advocate in handling health information, making a shared medical decision, performing multiple tasks related to the daily management of the disease, and keeping older adults at home. Cajavilca and Sadarangani [51] recently introduced the concept of dementia literacy, defined as the ability to acquire dementia-related knowledge to guide decision-making, identify gaps in caregiving support, and secure access to necessary resources for long-term care—all while maintaining relationships with an interdisciplinary team of healthcare professionals. They also discuss the relevance of dementia literacy for informal caregivers. From this perspective, higher HL skills are expected to enhance caregivers’ empowerment, thereby reducing their perceived burden. This interpretation aligns with the findings reported by Häikiö et al. [18] in Norway and is consistent with our results.

Our study showed that, via bivariate analysis, caregivers’ GHQ-12 scores were significantly higher (i) among females caregivers; (ii) among those with lower HL (as measured with the S-TOFHLA score in the overall sample, and by both the NVS and the S-TOFHLA among males); (iii) among caregivers performing a greater number of care tasks; and (iv) among those assisting care-recipients with greater clinical complexity (i.e., lower MMSE or higher NPI). Moreover, via multivariate analysis, our study showed that (i) when considering caregivers’ features only (Model 1), GHQ-12 scores were predicted by sex, S-TOFHLA score, and the number of tasks performed; (ii) when both caregivers’ and patients’ features were included (Model 2), GHQ-12 scores were predicted by sex, S-TOFHLA score, and NPI score. Finally, the association between HL and psychological stress appeared to be more pronounced among male caregivers.

This study has several limitations. First, refusals to participate were not recorded, so a potential selection bias cannot be excluded. However, participant enrollment was consecutive and unselected, which likely reduced this risk. Second, the date of the first diagnosis was not taken into account, and thus information regarding the duration of caregiving—which may affect caregivers’ stress—were not collected. Similarly, data on support received from others (i.e., secondary caregivers) were not collected. Finally, this study used a cross sectional design, so causality could not be assessed.

## 5. Conclusions

According to our findings, functional HL and higher caregiving commitment—reflected by the number of care tasks performed and the severity of the care-recipient’s condition—significantly affect caregivers’ stress. Moreover, although female caregivers reported higher overall stress, the association between HL and psychological distress appeared to be particularly pronounced among male caregivers. From a social and a public health perspective, these results highlight the need to advocate for policies and legislation that ensure dedicated resources to reduce the burden on informal caregivers [52]. According to our data, targeted interventions should be designed to enhance caregivers’ HL—particularly among men—by providing training and support tailored to the tasks they perform. Such initiatives should aim to preserve caregivers’ mental health and improve their ability to manage dementia-related behavioral and psychological symptoms in their family members, to whom they are deeply emotionally and affectionately connected [53,54,55,56].

## Figures and Tables

**Figure 1 healthcare-13-03064-f001:**
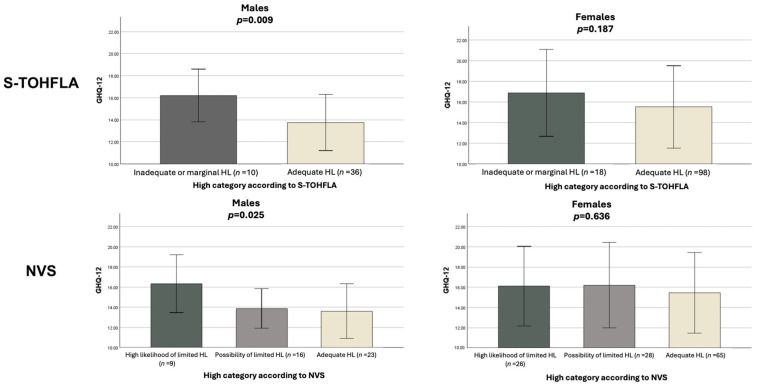
Association of HL according to S-TOHFLA and NVS with psychological stress. Analysis stratified by sex. Errors bars represent standard deviation. GHQ-12: General Health Questionnaire 12 items; HL: health literacy; S-TOHFLA: Short Form of the Test of Functional Health Literacy in Adults; NVS: Newest Vital Sign.

**Table 1 healthcare-13-03064-t001:** Demographic and clinical features of caregivers (*n* = 170).

Variables		Mean (SD)	Median (IQR)	*n* (%)	Missing Value
Age (years)		62.4 (11.7)	59 (54.5–71)	-	1
Sex	Female	-	-	120 (71%)	0
	Male	-	-	50 (29%)	
Kinship with care-recipient	Spouse	-	-	47 (28%)	0
	Offspring	-	-	123 (72%)	
Cohabitation with care-recipient		-	-	74 (44%)	2
Healthcare training		-	-	62 (37%)	4
Tasks provided	Personal hygiene	-	-	110 (65%)	1
	Dressing	-	-	95 (56%)	1
	Household chores	-	-	127 (75%)	1
	Feeding support	-	-	50 (29%)	1
	Mobility support	-	-	81 (48%)	1
	Surveillance	-	-	147 (89%)	1
	Drug administration	-	-	162 (90%)	1
	Number of tasks	4.5 (1.8)	5 (3–6)		1
Caregiving hours per week		74.7 (71)	30 (12–168)	-	8
Education (years)		12.2 (4.7)	13 (8–15)	-	3
GHQ-12	Score	15.3 (3.7)	15 (13–18)	-	1
	Score > 14			87 (48%)	1
NVS	Score	3.5 (2.1)	4 (2–6)	-	3
	High likelihood of limited HL			35 (21%)	
	Possibility of limited HL			44 (26%)	
	Adequate HL			88 (53%)	
S-TOFHLA	Score	83.4 (25.2)	94 (82.3–98)	-	8
	Inadequate or marginal HL			28 (17%)	
	Adequate HL			134 (83%)	

GHQ: General Health Questionnaire; HL: health literacy; NVS: Newest Vital Sign; S-TOFHLA: Short Form of the Test of Functional Health Literacy in Adults.

**Table 2 healthcare-13-03064-t002:** Caregivers’ features: comparison by sex.

Categorical Variables		Male (*n* = 50) *n* (% *)	Female (*n* = 120) *n* (% *)	Chi-Square	*p*
Kinship with care-recipient	Spouse Offspring	22 (44%) 28 (56%)	25 (20.8%) 95 (79.2%)	9.47	0.002
Cohabitation with care-recipient		25 (50%)	49 (40.8%)	1.02	0.312
NVS Score	High likelihood of limited HL Possibility of limited HL Adequate HL Missing value	9 (18%) 16 (32%) 23 (46%) 2 (4%)	26 (21.7%) 28 (23.3%) 65 (54.2%) 1 (0.8%)	1.67	0.792
S-TOFHLA	Inadequate or marginal HL Adequate HL Missing value	10 (20%) 36 (72%) 4 (8%)	18 (15%) 98 (81.7%) 4 (3.3%)	0.89	0.196
**Continuous Variables**		**Male** **Mean (SD)**	**Female** **Mean (SD)**	**t**	** *p* **
Age (years)		67.5 (12.2)	60.3 (10.9)	3.59	0.001
Education (years)		11.9 (5.1)	12.3 (4.6)	0.75	0.659
Caregiving hours per week		81.7 (74.9)	71.9 (69.5)	0.79	0.431
Number of Tasks Provided		4.1 (1.9)	4.7 (1.9)	2.06	0.041

* Column percentages; S-TOHFLA: Short Form of the Test of Functional Health Literacy in Adults; NVS: Newest Vital Sign; SD: standard deviation.

**Table 3 healthcare-13-03064-t003:** Association between caregivers’ features and psychological stress.

Categorical Variables		Mean GHQ-12 Score (SD)	Statistic	Effect Size (95% CI)	*p*
Sex	Female Male	15.8 (4.0) 14.2 (2.7)	t = 2.97	0.43 * (0.09, 0.76)	0.004
Kinship with care-recipient	Spouse Offspring	15.7 (3.7) 15.2 (3.7)	t = 0.80	0.14 * (−0.20, 0.48)	0.422
Cohabitation with care-recipient	Yes No	15.1 (3.7) 15.5 (3.8)	t = 0.67	0.11 * (−0.20, 0.41)	0.502
NVS Score	High likelihood of limited HL Possibility of limited HL Adequate HL	16.2 (3.7) 15.4 (3.7) 15.0 (3.8)	F = 1.28	0.02 ^ (0.00, 0.06)	0.280
S-TOFHLA	Inadequate or marginal HL Adequate HL	16.6 (3.6) 15.0 (3.7)	t = 2.07	0.43 * (0.02, 0.84)	0.040
**Continuous Variables**		**Correlation with GHQ-12 (Rho)**			** *p* **
Age (years)		0.047			0.545
Education (years)		−0.134			0.086
Caregiving hours per week		0.142			0.073
Number of Tasks Provided		0.192			0.013

GHQ-12: General Health Questionnaire 12 items; HL: Health Literacy; S-TOHFLA: Short Form of the Test of Functional Health Literacy in Adults; NVS: Newest Vital Sign. * Cohen’s d was reported (measure of effect size for two-group comparisons). ^ eta-squared (measure of effect size for the 3-group comparison).

**Table 4 healthcare-13-03064-t004:** Predictors of GHQ-12 score (linear regression models with backward deletion): including caregivers’ features only (Model 1); including both caregivers’ and patients’ features (Model 2). SE: Standard Error.

Model 1	B (SE)	*p*	Excluded Variables Using Backward Deletion
Female sex	1.32 (0.65)	0.043	caregiver age, caregiver education, caregiver kinship with care-recipient
S-TOFHLA °	−1.45 (0.78)	0.064	
Number of tasks provided *	0.31 (0.16)	0.060	
**Model 2**	**B (SE)**	** *p* **	**Excluded Variables Using Backward Deletion**
Female sex	1.72 (0.60)	0.004	caregiver age, caregiver education, caregiver kinship with care-recipient, number of tasks provided, care-recipient MMSE
S-TOFHLA °	−1.41 (0.71)	0.048	
NPI *	0.15 (0.03)	<0.001	

° adequate health literacy vs. inadequate or marginal health literacy. * continuous variable.

## Data Availability

Due to privacy and ethical concerns, the data that support the findings of this study are only available from the corresponding author upon reasonable request.

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
