# Peer review of "Gender Differences in the Relationship Between Health Literacy and Stress Among Caregivers of Older Adults with Dementia"

_healthcare, 2025, doi:10.3390/healthcare13233064_

Round 1

Reviewer 1 Report

Comments and Suggestions for Authors

This manuscript brings no novelty in regards to the previously published study on the same dataset (https://pmc.ncbi.nlm.nih.gov/articles/PMC9580430/#sec19).  There are methodological flaws - you use burden and stress as they were the same construct, you didn't measure burden with a standard instrument and there are errors in the whole text. The statistical analysis does not contain explanation of the presumptions of linear regression and are misinterpreted. Altogether, no new or relevant information and a lot of errors. 

Authorship statement: not all authors have fulfilled the ICMJE authorship criteria. There are people listed as authors that are not reported as been involved in writing the original draft or writing and editing and therefore I also have ethical concerns about this manuscript.  

Comments on the Quality of English Language

/

Author Response

Comment 1: This manuscript brings no novelty in regards to the previously published study on the same dataset (https://pmc.ncbi.nlm.nih.gov/articles/PMC9580430/#sec19). 

Response 1: The findings presented in this paper are derived from a broader research project encompassing multiple objectives and comprising both cross-sectional and longitudinal phases. During the cross-sectional phase, extensive data were collected on both caregivers and the older adults under their care. The first publication arising from this study reported part of these results, specifically examining “the prevalence of inadequate health literacy (HL) in a sample of informal family caregivers of older adults with dementia. As a secondary aim, we assessed the relationship between HL and the characteristics of caregivers and persons with dementia” (Lorini, 2022). Building upon those findings, the present paper focuses on caregivers’ stress and its determinants, with particular attention to health literacy as a predictive factor. In this context, we consider the current study to provide an original contribution that extends and complements the findings of the previous publication.

Comment 2: There are methodological flaws - you use burden and stress as they were the same construct, you didn't measure burden with a standard instrument and there are errors in the whole text.

Response 2: We thank the reviewer as this observation allows us to better define this point. We have changed the term “burden” into “stress” throughout the paper.

We chose to use a measure of psychological stress, such as the General Health Questionnaire, as measure of caregiver burden, such as the Relative Stress Scale or the Caregiver Burden Inventory, would not have been suitable to assess formal caregiver. Anyway, as older subjects without family caregivers were very few, we chose at the end to exclude them from the sample. The General Health Questionnaire is widely used in caregiver literature as a measure of caregiver stress, and has been found to be strongly correlated with caregiver burden: see for reference the recent large study by Van Droogenbroeck F et al., Informal caregiving and mental health: results from the Belgian health interview survey 2013 and 2018. BMC Public Health, 2025

Comment 3: The statistical analysis does not contain explanation of the presumptions of linear regression and are misinterpreted.

Response 3: We thank the reviewer of raising the important methodological point. We formally tested the assumptions of linear regression, including normality distribution of residuals, lack of significant collinearity of covariates and homoscedasticity across the values of independent variables. We confirmed the normal distribution of residuals and the lack of collinearity, while we found that the assumption of homoscedasticity was not fulfilled for model 1, while it was confirmed for model 2. Therefore, we conclude that the estimates obtained may not be stable enough in model 1, while model 2 produces robust estimates of stress across all predicted values. P-P plots and scatterplot of residuals of the two models are included in Supplementary Figure.

We tried to clarify the interpretation of results and hope this is clear now.

Text added in Method section

Regression assumptions were tested as follows: normality of residuals was verified using the P-P plots, collinearity was assessed using the Variance Inflation Factor (VIF), with VIF <5 indicating acceptable levels, and homoscedasticity was evaluated by inspecting scatterplots of standardized residuals by standardized predicted values and confirmed with the Breusch-Pagan test.”

 Statements added among Results:

“However, due to the lack of homoscedasticity, the estimates of Model 1 may not be considered stable (see Supplementary Figure 1)”.

“Model 2 produced robust estimates of stress across all predicted values, as shown by homoscedasticity assessment (see Supplementary Figure)

Comment 4: Altogether, no new or relevant information and a lot of errors. 

Response 4: The role of heath literacy as predictor of caregiver stress has been rarely studied in dementia literature. The exploratory analysis regarding different association between health literacy and caregiver stress in different genders provide new results. Such results should be confirmed in different samples and their relevance in a public health and clinical perspective are obviously to be confirmed in larger, possibly intervention studies.

We conducted a thorough check of the manuscript to remove language errors and clarify possible ambiguities. If specific errors are still present, we would be grateful if you may report them.

Comment 5: Authorship statement: not all authors have fulfilled the ICMJE authorship criteria. There are people listed as authors that are not reported as been involved in writing the original draft or writing and editing and therefore I also have ethical concerns about this manuscript.  

Response 5: We thank the reviewer for this comment. We confirm that all authors have been actively involved in the preparation of the manuscript, including the original drafting, writing, and editing processes. We have revised the text accordingly to clarify this point.

Reviewer 2 Report

Comments and Suggestions for Authors

The manuscript is solid and addresses a highly relevant topic with clear social and public health implications. However, before final acceptance, the following should be addressed:

  1. While GHQ-12, NVS, and S-TOFHLA are described, other instruments such as the Mini-Mental State Examination (MMSE) and the Neuropsychiatric Inventory (NPI) are only mentioned in the results but not properly introduced in the Materials and Methods section. The psychometric properties (validity, reliability, adaptation to the Italian context) of all instruments used in the study should be explicitly reported to strengthen methodological rigor.

  2. The manuscript states that the sample size was estimated based on an expected prevalence of low HL, a 95% confidence interval, and a 5% margin of error. However, it does not specify the expected prevalence value used in the calculation nor the formula applied. This should be clarified to justify the target sample size of 180 caregivers.
  3. The sample description (age, sex, education, tasks performed, etc.) is currently included in the Results section but should instead be moved to Methods (under a subsection such as Sample description). The Results section should be reserved for analytical findings.

  4. It is unclear whether tests of normality (e.g., Kolmogorov–Smirnov, Shapiro–Wilk) and homoscedasticity (e.g., Levene’s test) were performed to justify the use of parametric tests such as t-test and ANOVA.
  5. Missing data are reported in the tables, but the manuscript does not specify how missing values were handled: were they excluded by listwise deletion, imputed, or managed per analysis? This must be clarified in Methods.
  6. In the results, means, standard deviations, and p-values are presented, but the actual test statistics (t, F, χ², Rho, etc.) are missing. Reporting these is necessary for transparency.
  7. Effect sizes (e.g., Cohen’s d, η², adjusted R²) are not reported, which would allow readers to assess the magnitude of the associations, not only their statistical significance.
  8. The figures could be improved to meet scientific publication standards (clearer typography, adjusted scales, complete legends). It is also important to indicate whether the error bars represent standard deviations (SD) or standard errors of the mean (SEM), as the values shown appear large and may mislead readers.

Author Response

Comment 1: While GHQ-12, NVS, and S-TOFHLA are described, other instruments such as the Mini-Mental State Examination (MMSE) and the Neuropsychiatric Inventory (NPI) are only mentioned in the results but not properly introduced in the Materials and Methods section. The psychometric properties (validity, reliability, adaptation to the Italian context) of all instruments used in the study should be explicitly reported to strengthen methodological rigor.

Response 1: We thank the reviewer for this comment. A brief description of the MMSE and of the NPI was added, including references. Moreover, psychometric proprieties or details on the validation studies have been added.

Comment 2: The manuscript states that the sample size was estimated based on an expected prevalence of low HL, a 95% confidence interval, and a 5% margin of error. However, it does not specify the expected prevalence value used in the calculation nor the formula applied. This should be clarified to justify the target sample size of 180 caregivers.

Response 2: The sample size was calculated with respect to the primary outcome of the general research project, which was the percentage of caregivers with a low level of health literacy, measured using the two instruments was employed (S-TOHFLA and NVS), and was reported in a previous paper (Lorini C et al., Health literacy of informal caregivers of older adults with dementia: results from a cross-sectional study conducted in Florence (Italy). Aging Clin Exp Res, 2023). The formula we have used is the one to calculate the prevalence of a condition or a disease, namely:

N= Z2x P (1-P)

d2

where:

?: the Z-score that corresponds to the desired confidence level;

?: the expected prevalence (or proportion) of the condition in the population. 

?: the desired margin of error (or precision) for the estimate. 

At the beginning of the study, no Italian studies with the same objective have used the S-TOHFLA as a tool for measuring health literacy. For this reason, the sample size calculation was based on the results of studies from the literature (Garcia, 2013; Lindquist, 2011), which report a percentage of caregivers with “inadequate health literacy” of approximately 30–36%. Considering the aim of calculating a 95% confidence interval with a precision of 0.05, the planned sample size is 180 participants.

With regard to the NVS, the results of a study conducted in Tuscany to measure the level of health literacy among a sample of formal caregivers reported a prevalence of low health literacy of about 40% (Bonaccorsi, 2016). Again, considering a 95% confidence interval with a precision of 0.05, the planned sample size is approximately 188 participants.

These information have been added in the Method section.

Comment 3: The sample description (age, sex, education, tasks performed, etc.) is currently included in the Results section but should instead be moved to Methods (under a subsection such as Sample description). The Results section should be reserved for analytical findings.

Response 3: The 3.1 section (Sample description) has been moved to the Methods section, as suggested.

Comment 4: It is unclear whether tests of normality (e.g., Kolmogorov–Smirnov, Shapiro–Wilk) and homoscedasticity (e.g., Levene’s test) were performed to justify the use of parametric tests such as t-test and ANOVA.

Response 4: We formally performed normality distribution of residuals with P-P plots and homoscedasticity tests for t-tests and ANOVA measures. This statement was added among Methods

Comment 5: Missing data are reported in the tables, but the manuscript does not specify how missing values were handled: were they excluded by listwise deletion, imputed, or managed per analysis? This must be clarified in Methods.

Response 5: Missing values were excluded by listwise deletion. This information has been added in the method section.

Comment 6: In the results, means, standard deviations, and p-values are presented, but the actual test statistics (t, F, χ², Rho, etc.) are missing. Reporting these is necessary for transparency.

Response 6: Pearson Chi-square, Student and F values were added in the Tables. Rho was already reported.

Comment 7: Effect sizes (e.g., Cohen’s d, η², adjusted R²) are not reported, which would allow readers to assess the magnitude of the associations, not only their statistical significance.

Response 7: We thank the reviewer for this suggestion to improve the paper report. We have updated the Methods accordingly and we ha added effect size in Table 3 and in the text describing the exploratory analysis of analysis stratified for caregiver sex.”

Text added among Methods:

“Effect size of means’ comparisons were estimated with Cohen’s d for Student t and with eta-squared for ANOVA”.

Text amended among Results:

“In the exploratory analysis stratified for caregiver sex, a significant association emerged between HL and GHQ-12 score among males, as lower GHQ-12 score were associated with higher HL, whether considering the 3-level NVS [eta squared 0.15 (95% CI 0.00, 0.32)] or the S-TOFHLA [eta-squared 0.14 (95% CI 0.01, 0.33), corresponding to a medium-to large effect size]. Conversely, for female caregivers no significant association emerged between HL and GHQ-12 score (Figure 1).”

Comment 8: The figures could be improved to meet scientific publication standards (clearer typography, adjusted scales, complete legends). It is also important to indicate whether the error bars represent standard deviations (SD) or standard errors of the mean (SEM), as the values shown appear large and may mislead readers.

Response 8: The resolution of the Figures was increased to 300 dpi. The same scales are reported on y-axis of all graphics in Figure 1. All abbreviations are explained in the caption. Error bars represent standard deviation

Reviewer 3 Report

Comments and Suggestions for Authors

The manuscript needs revision as described below. Overall, it needs a better flow for readers to follow the objective. 

1- Irrelevant Title. Please reconsider. Additionally, I am unsure why the title has a question mark.

2- Background/ Objectives- There is no background for the  

3- The abbreviation in line 23, the first line in the abstract, “HL”, should be spelled out for the first time. It is spelled in line 70, but it needs to be at the beginning.

4- The term 'older people' is not scientifically accurate in describing this cohort, which refers to individuals aged 65 and above. It should be addressed as older adults, individuals 65 and older, or the geriatric population. Again, line 113, ‘elderly demented people’, wrong terminology used.

5- Newest Vital Sign (NVS). Does that mean the latest / the most recent vital sign?

6- Is the term ‘offspring’ used to describe adult children, like sons and daughters, or extended to grandchildren as well?

Comments on the Quality of English Language

The manuscript lacks a seamless flow, making it difficult to read and follow. I think that an English-speaking reviewer should review the manuscript before resubmitting. 

Author Response

Comment 1: Irrelevant Title. Please reconsider. Additionally, I am unsure why the title has a question mark.

Response 1: The title was modified to be more relevant (Gender Differences in the Relationship Between Health Literacy and Stress Among Caregivers of Older Adults with Dementia)

Comment 2: Background/ Objectives- There is no background for the  

Response 2: Unfortunately, we can not understand your request. Could you please clarity?

Comment 3: The abbreviation in line 23, the first line in the abstract, “HL”, should be spelled out for the first time. It is spelled in line 70, but it needs to be at the beginning.

Response 3: The terms “health literacy” have been added in line 23.

Comment 4: The term 'older people' is not scientifically accurate in describing this cohort, which refers to individuals aged 65 and above. It should be addressed as older adults, individuals 65 and older, or the geriatric population. Again, line 113, ‘elderly demented people’, wrong terminology used.

Response 4: The term “older people” has been changed in “older adults”; “elderly demented people” has been changed in “older adults with dementia”

Comment 5: Newest Vital Sign (NVS). Does that mean the latest / the most recent vital sign?

Response 5: The Newest Vital Sign (NVS) is a questionnaire to measure health literacy. It was developed in English in the USA, and then validated in many other languages and for many other populations.

Comment 6: Is the term ‘offspring’ used to describe adult children, like sons and daughters, or extended to grandchildren as well?

Response 6: The term “offspring” is used to describe sons and daughters.

Comment 7: Comments on the Quality of English Language

The manuscript lacks a seamless flow, making it difficult to read and follow. I think that an English-speaking reviewer should review the manuscript before resubmitting. 

Response 7: The manuscript has been reviewed to improve language and clarity